# Analysis of Crash Frequency and Crash Severity in Thailand: Hierarchical Structure Models Approach

**Thanapong Champahom [1], Sajjakaj Jomnonkwao [2], Chinnakrit Banyong [2], Watanya Nambulee [3], Ampol Karoonsoontawong [4] and Vatanavongs Ratanavaraha [2,*]**

1 Department of Management, Faculty of Business Administration, Rajamangala University of Technology Isan, Nakhon Ratchasima 30000, Thailand; thanapong.ch@rmuti.ac.th
2 School of Transportation Engineering, Institute of Engineering, Suranaree University of Technology, Nakhon Ratchasima 30000, Thailand; sajjakaj@g.sut.ac.th (S.J.); m6040185@g.sut.ac.th (C.B.)
3 Division of Logistic, Engineering Faculty of Engineering, Nakhon Phanom University, Nakhon Phanom 48000, Thailand; nambulee.w@gmail.com
4 Department of Civil Engineering, Faculty of Engineering, King Mongkut's University of Technology Thonburi, Bangkok 10140, Thailand; ampol.kar@kmutt.ac.th
* Correspondence: vatanavongs@g.sut.ac.th; Tel.: +66-0-4422-4238

**Abstract:** Currently, research on the development of crash models in terms of crash frequency on road segments and crash severity applies the principles of spatial analysis and heterogeneity due to the methods' suitability compared with traditional models. This study focuses on crash severity and frequency in Thailand. Moreover, this study aims to understand crash frequency and fatality. The result of the intra-class correlation coefficient found that the spatial approach should analyze the data. The crash frequency model's best fit is a spatial zero-inflated negative binomial model (SZINB). The results of the random parameters of SZINB are insignificant, except for the intercept. The crash frequency model's significant variables include the length of the segment and average annual traffic volume for the fixed parameters. Conversely, the study finds that the best fit model of crash severity is a logistic regression with spatial correlations. The variances of random effect are significant such as the intersection, sideswipe crash, and head-on crash. Meanwhile, the fixed-effect variables significant to fatality risk include motorcycles, gender, non-use of safety equipment, and nighttime collision. The paper proposes a policy applicable to agencies responsible for driver training, law enforcement, and those involved in crash-reduction campaigns.

**Keywords:** spatial model; zero-inflated; crash severity; hierarchical structure analysis; Thai highways

## 1. Introduction

### 1.1. Background

The Thailand Road Safety Center [1] stated that the reduction of road crashes is a crucial component of the country's sustainable development. Therefore, the Thai government intends to reduce annual crash fatality rates to less than 10 people for every 100,000 people by 2022. However, according to the WHO estimates, the annual death rate in Thailand remains at 32.7 people for every 100,000 people. Thailand is ranked 9th in the world in terms of crash fatality rates [2].

Given the statistical trend of road crashes under the responsibility of Thailand's Ministry of Transport, previous studies found that the highest number of crashes and deaths occurred on roads under areas supervised by the Department of Highways (DOH). Specifically, 17,045 crashes, 2651 deaths, and 15,835 injuries occurred in 2018 (2562 B.E.) [3]. Highways are main roads allocated for the transportation of people or goods. It leads to relatively high averages in daily traffic volume. Thus, a decrease in the said rates should be initiated on highways.

The significance of the frequency of road traffic crashes requires an analysis of several characteristics based on the unique features of each area. These include road designs [4],

environment, and driver characteristics, which can potentially result in different driving behaviors [5].

*1.2. Crash Modeling with Spatial Approach*

The models predicting the relationship between the factors of crash frequency and crash severity typically applied traditional approaches (i.e., used fixed parameters) in the first phase [6–9]. A recent stream of research on traffic crashes considered an analysis of crash data on the basis of a multilevel or hierarchical model. For example, Dupont et al. [10], Huang and Abdel-Aty [11] indicated that traditional crash prediction models, such as regression models, can be taken as a multilevel structure. After the review, the authors concluded that cross-group heterogeneity and spatiotemporal correlation may be vital and may potentially occur on crash data. For example, it may apply to studies that analyzed crash frequencies at the road segment level [12–14]. Ziakopoulos and Yannis [15] made a summary of a recent study reviewing spatial approaches in road safety. They concluded that because transportation involved a specific distance of each area, the consideration of road crash analysis should be by spatial analysis. This means an attempt to test spatial heterogeneity. The spatial heterogeneity model analysis refers to determining the relationship between random parameters and observed events to be spatially changed. Therefore, this study suggests the application of a spatial analysis or a specific multilevel structure to achieve accurate and precise predictions.

Previous studies applied a multilevel data structure or spatial analysis, which required many tasks. Recent spatial studies include machine learning models. For example, Bao et al. [16] employed spatiotemporal deep learning to predict short-term risks by using the differences in grid sizes in their analysis. Cai et al. [17] studied the Bayesian-integrated spatial crash frequency model, which stratified the data into macro- and micro-levels based on the differences of road entities such as segments and intersections. The results of the study showed that black spots of both levels could be identified.

Lenguerrand et al. [18] applied the Monte Carlo methods to observe and analyze crash severity and compared three models: the multilevel logistic modeling (MLM), the generalized estimating equation, and the logistic model. They concluded that the MLM predicted crash severity with the highest accuracy. Kim et al. [19] conducted a study on the severity of crash injury levels by applying hierarchical binomial logistic models. They specified that the lower level pertains to the collision type, whereas the higher level refers to the physical characteristics of an intersection. They found a total of 9–50% variance corresponding to estimated intra-class correlations. This indicates the proportion of the total variance for different crash types that can be attributed to intersection characteristics. Moreover, they compared the coefficients with the MLM and the non-MLM (NMLM) models. Here, they found that the MLM potentially displayed potentially more accurate predictions. This finding is consistent with those of Huang et al. [20], Chen et al. [21], Chen et al. [22].

For studies defining area as a second-level variable, Yannis et al. [23] applied a negative binomial (NB) multilevel modeling with spatial analysis. The authors used spatial variability at the county level to analyze the frequency of drinking and driving in cities in Greece. Park et al. [24] investigated crash severity involving commercial motor vehicle crashes by specifying the first level as the crash level and the second level as the region level or zoning. Zoning uses zip codes as a separator. In other cases, Lee et al. [25] found that the model divided by a zip code tabulation area produced the best results. Jones et al. [4] explored the individual characteristics of collisions in relation to fatality risks. They applied the multilevel regression model to analyze the impact of multilevel models mentioned above. The final model was divided into three levels: level 1, driver characteristics; level 2, crash type; and level 3, area/municipality characteristics. Building on the final model, Ahmed et al. [26] applied such model for cases observed on mountain roads. On the other hand, Flask et al. [13] used it to investigate single-vehicle crashes, such as motorcycles [27], and corresponding crash frequency models [28]. Undoubtedly, this technique is extensively

applied and proven to produce more accurate predictions than the traditional models. In another study focusing on the exposure of the parameter-influencing factors through meta regressions, Ziakopoulos and Yannis [29] conducted a spatial analysis and found that traffic volume positively contributed to the number of crashes from the first set of variables. For roadway length dimension, an increase in roadway length tended to increase the number of fatal crashes. Moreover, the variable of distance traveled would cause the crash count to increase only at the country level, as opposed to the traffic analysis zone.

### 1.3. Highway Organization of Thailand

The present study considers the organization of the Department of Highways (DOH) in Thailand. Here, the hierarchy of road responsibility is an interesting issue. Under the DOH, the "Office of Highways" is composed of 18 offices across the country (Figure 1), followed by "highway districts" composed of 106 districts. The different characteristics of the two sub-organizations, such as the congestion index, scope of responsibility, annual traffic volume, responsibility for road distance, and annual budget, may affect crashes and their severity levels. This notion seeks to establish a spatial model by determining the two levels of factors that influence the dependent variables: crash case level (i.e., drivers, environment, or roads) and spatial level. They pertain to the operational area (i.e., operational area under responsibility, traffic volume, and congestion index).

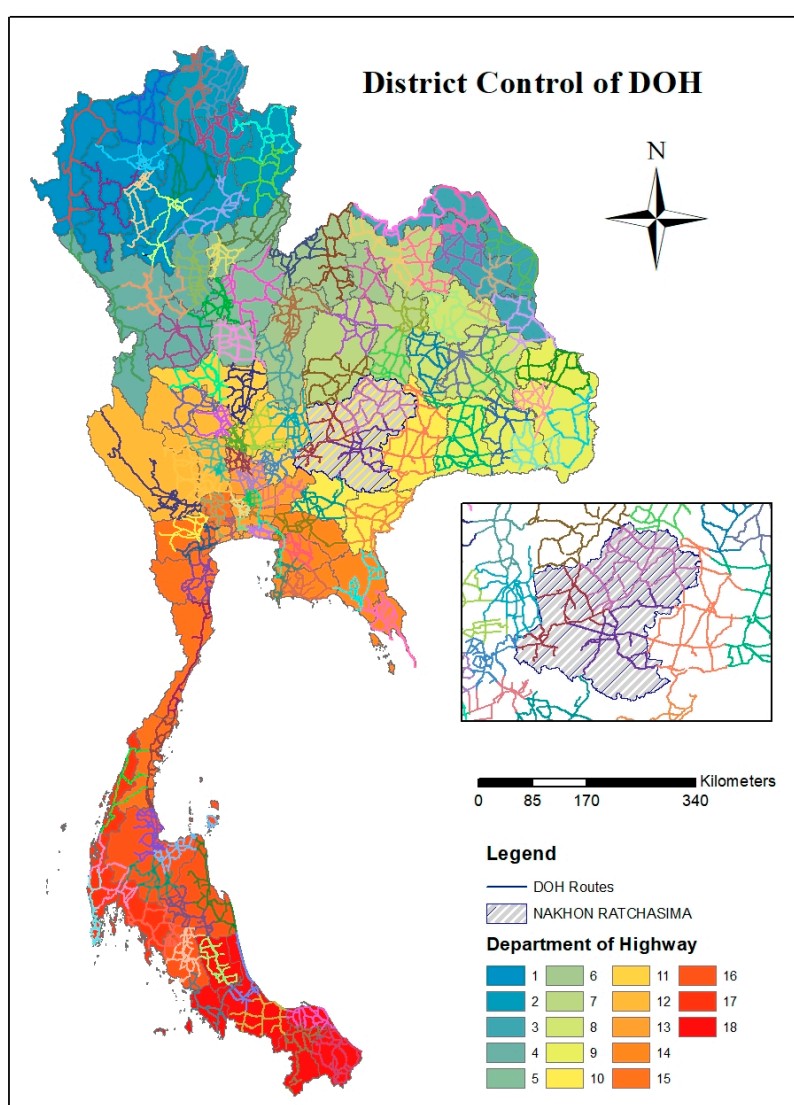

**Figure 1.** Responsibility area of the Department of Highway offices in Thailand.

Ziakopoulos et al. [15] performed a review where they found that there was no spatial approach on a middle-income and developing country. They also suggested that the heterogeneity of crashes may be different. They are expected to produce varying results due to varying driving culture or other unobserved factors. As such, this study could fulfill the scientific knowledge of road traffic crashes in developing countries. Furthermore, if the model results are attained, they can be used to determine spatial policies. The results can likewise promote campaigns for reducing the number of crashes and death rates in Thailand, such as in other middle-income and developing countries.

The rest of the paper is structured as follows: Section 2 describes the methods used for data analysis; Section 3 presents the results of the analysis and a discussion in two parts: the crash frequency model and crash severity model; and Section 4 provides the summary and limitations of the study.

## 2. Methods

Studies on road traffic crash analysis at the local and global scales are divided into two groups of factors based on outcome: (1) crash occurrences and (2) crash fatalities. Figure 2 presents the procedure followed in this study.

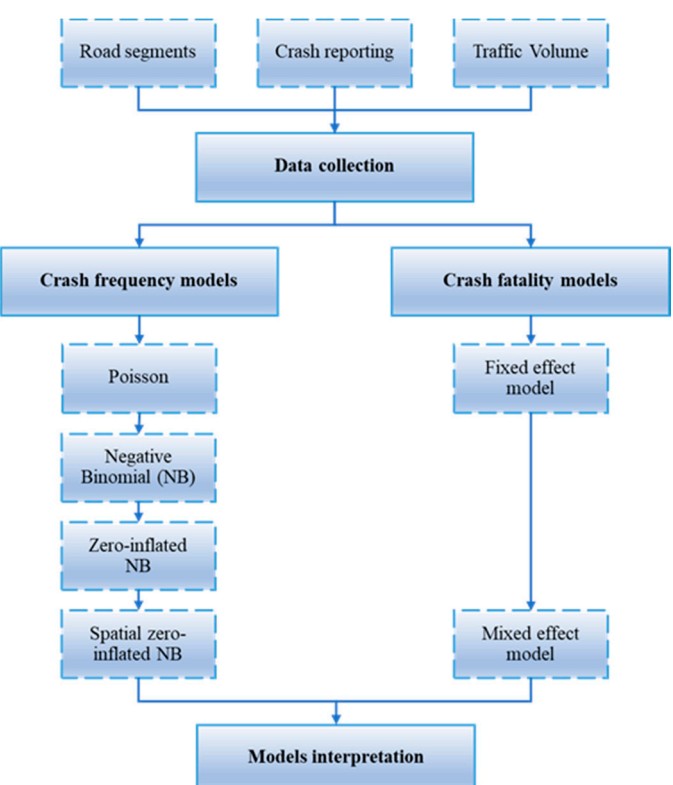

**Figure 2.** Procedure.

### 2.1. Crash Frequency Models

Count data consist of non-negative integer values. They are normally used for transportation models, such as number of routes, drivers' change per week, and number of crashes on roads per year. For the probability equation of the Poisson model, $\alpha$ takes a value of zero. This implies that the choice between the two models (i.e., Poisson and NB) is dependent on the value of $\alpha$, which is relatively frequently over-dispersed model. Washington, et al. [30] provided a test for overdispersion. If the overdispersion parameter is significant, the NB model would be more appropriate than the Poisson model.

The probability of the number of collisions occurring on a road segment is considered an NB distribution. Predicting the annual number of crashes per year should consider

that no crashes may take place on certain roads. Thus, this variable can be categorized as normal-count and zero-count (Figure 3).

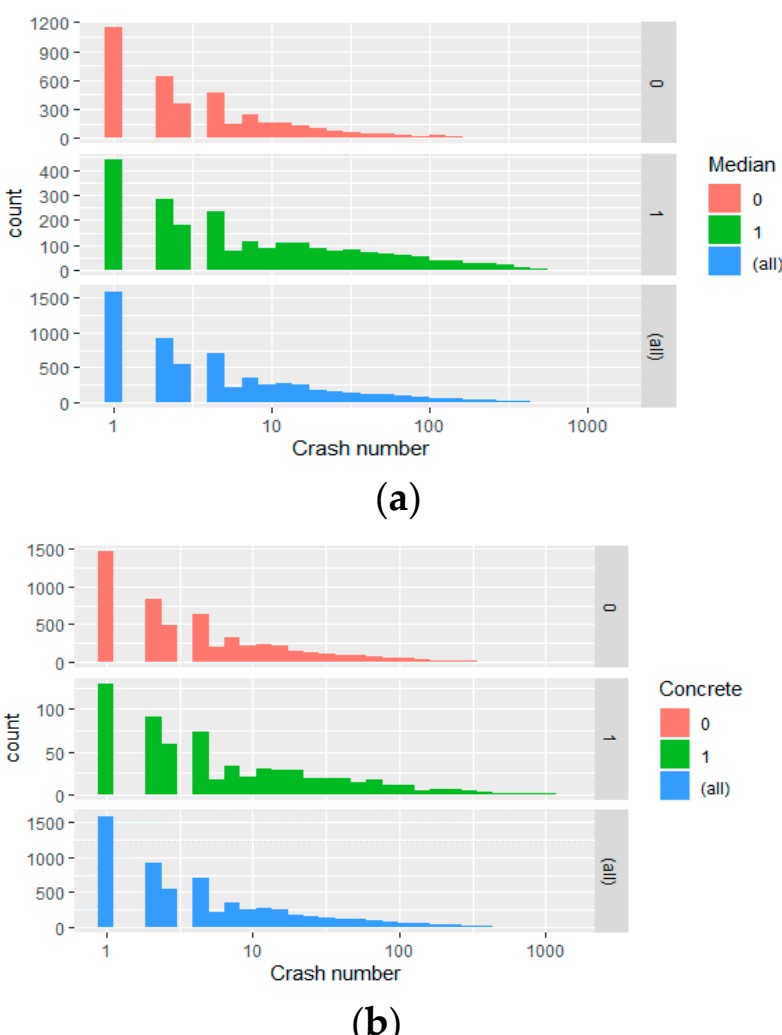

(a)

(b)

**Figure 3.** Distribution of crash occurrences: Horizontal axes in both were log−10 scale: (**a**) The distribution was divided into road and undivided road. (**b**) The distribution was divided by the pavement type.

The general model may not be comprehensive for separating the analysis into two parts. Therefore, the most suitable model for the dual state is the zero-inflated model. It is called the zero-inflated negative binomial (ZINB) when established on the NB model [31]. Vuong statistics were used to compare between the ZINB and the NB model with $f_1(.)$ as the density function of the ZINB and $f_1(.)$ as the density of function of the NB model. The decision guidance suggested that if the Vuong statistic is 1.96, and the $t$-statistics of the NB overdispersion parameter is $\alpha > |1.96|$, the zero-inflated models are plausible. This is because it is evident that some simultaneous zero-altered process is at work [32]. "Part III: Count and discrete dependent variable models" of Washington, et al. [30] further discusses this analysis. The models have crash frequency as the response variable and the highway characteristics or traffic volume as explanatory variables.

In certain cases, the reasons for the prediction may include a mutual correlation among observable data. This relationship is due to spatial data, such as data on crashes within the same area. Thus, the effect should be determined and not just observed. As such, the present study classifies roads into areas according to the designation of the sub-departments of the Department of Highway (DOH). In light of this relationship, model

application should generate random and fixed effects, especially when an unpredictable factor is taken as an indicator variable.

The equation for the random effects is modified using the equation for log-linear function:

$$Y_{ij} \sim \text{Poisson}(\lambda_{ij}) \tag{1}$$

$$\log \lambda_{ij} = \beta_{0j} + \sum_{m=1}^{M} X_{ijm}\beta_{mj} + \varepsilon_{ij} \tag{2}$$

$$\beta_{0j} = \gamma_{00} + \mu_{0j} \tag{3}$$

$$\beta_{mj} = \gamma_{m0} + \mu_{mj} \tag{4}$$

In this equation, $\lambda_{ij}$ refers to the expected number of crashes for road segment $i$ in group $j$ (i.e., the regulatory space of the sub-departments of the DOH, where unobserved heterogeneity is expected). On the other hand, $X_{ijm}$ pertains to the vector of the explanatory variable $m$th, while $\beta_{mj}$ stands for the parameter prediction vector. $\varepsilon_i$ is the random effect representing within-area variance. $\mu_{0j}$ and $\mu_{mj}$ denote the unique effect associated with sub-areas of the DOH. $\varepsilon_i$, $\mu_{0j}$, and $\mu_{mj}$ are usually assumed at normal distribution with the mean zero. Spatial correlation is calculated based on the proportion of spatial variation from the total variation [33–36]. The equation is as follows:

$$\text{Spatial correlation} = \frac{\sigma_\eta^2}{\sigma_\eta^2 + \sigma_e^2} \tag{5}$$

where $\sigma_\eta^2$ represents the variance obtained from the prediction of $i$ or variance within the same sub-DOH area and $\sigma_e^2$ refers to the variance obtained from the estimation of the fixed effects or the variance between the sub-areas of the DOH.

Program R was used for analysis. The dplyr package (a grammar of data manipulation) was used for data input [37]. The ggplot2 package was used to create the graphic of the data distribution. The traditional crash frequency models include Poisson that was analyzed by glm (stats) (fitting generalized linear models) and NB that was analyzed by glm.nb (MASS) (fit a NB generalized linear model). On the other hand, the ZINB was analyzed by zeroinfl (pscl) (fit zero-inflated regression models for count data via maximum likelihood) [38]. The Vuong (pscl) was used to check the plausibility of the zero state. With regard to the final model, the spatial zero-inflated negative binomial models (SZINB) were analyzed by glmmTMB (fit a generalized linear mixed model using Template Model Builder) [39].

### 2.2. Crash Severity Modeling

The study determined the factors influencing crash severity at the second level in terms of spatial characteristics. Such was carried out under the supervision of each district divided by the boundary of the province or district for large provinces. Logistic models applying spatial concepts are first used to determine driver factors (driver $i$) in the district group $j$. Each crash consisted of the driver ($i$th driver) in the $j$th road segment. The fatality of common drivers was calculated by using the log odds in general areas. In other words, various characteristics of the areas have indifferently affected the likelihood of fatality, which determines the two severity levels including the fatal and non−fatal crashes.

$$Y_{ij} = \begin{cases} 1; & \text{if case of fatal crash} \\ 0; & \text{if case of non-fatal crash} \end{cases}$$

Here, $Y_{ij}|p_{ij} \sim Bernouilli(p_{ij})$, $p_{ij} = \text{Pr}(Y_{ij} = 1)$ represents the probability of driver $i$ from road $j$ that has fatal crashes, coming from the relationship of estimated parameters of explanatory variables such as driver factors (e.g., gender, drunkenness, vehicle type),

road factors (e.g., median type, slope), and environmental factors (e.g., visibility). Such a relationship can be derived as follows:

$$logit\left(P_{ij}\right) = \eta_{ij} = \beta_{0j} + \sum_{k=1}^{K} \beta_{kj} X_{kij} \tag{6}$$

Here, $\eta_{ij}$ represents the log odds of driver $i$ on road $j$ that has fatal crashes. $\beta_{0j}$ is the constant or the average value of log odds 1 that has fatal crashes on road $j$ only. $X_{kij}$ is the *kth* explanatory variable at the individual level for predicting the odds of fatality $j$. On the other hand, $\beta_{kj}$ are parameter values that indicate the relationship (slope) between the driver level variables and the log odds that possibly causes the fatalities from the collisions. Equation (6) assumes that crashes on each road result in different degrees of severity. Thus, the equations are different from those of common logistic models. Each road has a constant value $\left(\beta_{0j}\right)$. However, the slopes $\left(\beta_{kj}\right)$ are different.

Considering a spatial model as an explanation for the variation in the regression coefficient and variables at the upper level based on road characteristics, these qualities are assumed to be constant. Slope values, such as the median types and intersections, exert their effects at the driver level (Equation (6)). These can vary according to the explanatory variables of each road segment $\left(Z_{mj}\right)$. The equations are formulated as follows:

$$\beta_{0j} = \gamma_{00} + \sum_{m=1}^{M} \gamma_{0m} Z_{mj} + \mu_{0j} \tag{7}$$

$$\beta_{kj} = \gamma_{k0} + \sum_{m=1}^{M} \gamma_{km} Z_{mj} + \mu_{kj} \tag{8}$$

$\gamma_{00}$ indicates the log odds of fatal crashes on the road. On the other hand, $Z_{mj}$ represents the $m$th explanatory variable for road $j$, and $M$ denotes the number of road level variables. $\gamma_{0m}$ indicate the slope or relationship with $Z_{mj}$ corresponding to $\beta_{0j}$ and $\beta_{kj}$, respectively. $\gamma_{00}$ is the intercept for or $\beta_{0j}$ and $\gamma_{k0}$ is the intercept for $\beta_{kj}$. $\mu_{0j}$ and $\mu_{kj}$ stand for the prediction errors in information at the road level and demonstrate the unique effect of road $j$. Kim et al. [19], Huang et al. [20], Xiao et al. [40], Se et al. [41], Champahom et al. [42] provide for extended details on the calculation presented above.

Before the spatial analysis, the test results should be obtained from the models without parameter estimation (unconditional model). This may be done by considering the proportion of variance of dependent variables (outcome) within (crash on the same road) and between (each road) groups. The latter are called intra-class correlation coefficients (ICC). They were calculated by assuming a logistic distribution for the dependent variables. For errors at the driver level with a variance value of $\pi^2/3$ [43], the ICC values can be calculated as follows:

$$\rho = \frac{\sigma_{\mu_0}^2}{\sigma_{\mu_0}^2 + \pi^2/3} \tag{9}$$

where $\sigma_{\mu_0}^2$ is the variance of the dependent variables between roads (level 2). The ICC values calculated for multilevel modeling should not be zero. In the case where the ICC value is zero, then no variation exists between the data of crashes on each road. According to previous research, the ICC values for second-level variables, such as intersection, should be higher than 0.09 or 9% [19].

The following step assesses the model suitability by comparing the intercept-only model with all parameters set to zero. Moreover, the convergence model is compared with parameter vector $\beta$. The values used for comparison were the log-likelihood of both models. This is also called pseudo-$R^2$ or McFadden's $\rho^2$:

$$\rho^2 = 1 - \frac{LL(\beta)}{LL(0)} \tag{10}$$

where $LL(\beta)$ is the log-likelihood value of the model with parameter estimation. On the other hand, $LL(0)$ represents that without or with parameter estimation but only with a constant value. If $\rho^2$ has a value close to 1, then the model can predict values close to the actual data [30].

R program was used for analysis. Moreover, glm (stats) with the family is binomial, and it was used to analyze the traditional model. The glmer (lme4) (fitting generalized linear mixed-effect models) was used for coefficients using the spatial analysis technique [44].

### 2.3. Data Collection

The highway crash reports on cases of road crashes from 2011 to 2017 that took place on Thai highways. During collisions, a highway officer investigates and records each case. The record contains information about the scene of the crash (i.e., specific name of the road and location in kilometers), date, physical characteristics of the road (i.e., median type and intersection), environmental factors (i.e., weather, time, and lighting), vehicle type, and crash type. The data collection was supported by the highway Bureau of Highway Safety, Department. Before developing the model, the correlations between explanatory variables were checked. It was found that there is no pair with correlation greater than 0.7, which was an appropriate indicator when collinearity begins to severely distort model estimation and subsequent prediction [45].

### 3. Results and Discussion

### 3.1. Crash Frequency Model

Table 1 provides a summary of the crash frequency models concerning road segment data. These were divided according to the physical characteristics of roads. The study found that the mean value of crash occurrence is 6.39 times, the segment length is 3.08 km, the lane width is 3.47 m, and the road with median is 33%. In terms of traffic volume, findings show that an average of 14,465 vehicles per day are on the streets, of which 16.33% are trucks. The maximum annual average daily traffic (AADT) is 339,248 vehicles per day. This value refers to the district of Thanya Buri and Klongluang of Prathum Tani province. This route is road no. 1 (Phahonyothin Road), which has 10–14 lanes with a barrier in the middle. This area is an economic and complex area. This means that there are many trip production and trip attractions including industry zones, complex residences, universities, and shopping malls. In addition, the road connects the biggest regions of Thailand including the "North" and the "Northeast." The official report of the AADT on this road could be assessed by the Thailand Department of Highways [46].

**Table 1.** Data description of the crash frequency model.

| | Description | N | M | SD | MIN | MAX |
|---|---|---|---|---|---|---|
| Crash number | Crash number | 16,933 | 6.39 | 33.89 | 0 | 1329 |
| Length | Length of segment | 16,933 | 3.08 | 5.02 | 0 | 63.16 |
| No. of Lane | Number of lanes | 16,933 | 3.19 | 1.78 | 1 | 14 |
| Concrete * | Pavement type (1 = Concrete, 0 = Otherwise) | 16,933 | 0.10 | 0.3 | 1 | 2 |
| Lane width | Lane width (m) | 16,933 | 3.47 | 0.21 | 2.5 | 6 |
| Footpath * | Shoulder type (1 = Footpath, 0 = otherwise) | 16,933 | 0.05 | 0.21 | 1 | 2 |
| Shoulder width | Shoulder width (m) | 16,933 | 1.74 | 0.88 | 0 | 7.2 |
| Median * | Divided road (1 = Yes, 0 = Otherwise) | 16,933 | 0.33 | 0.47 | 1 | 2 |
| Median width | Median width (m) | 16,933 | 0.84 | 3.27 | 0 | 15 |
| AADT | Average annual traffic volume (vehicle) | 16,933 | 14,465.51 | 24,286.45 | 58 | 339,248 |
| Percent of truck | Percentage of truck volume | 16,933 | 16.33 | 11.77 | 0 | 72.51 |
| LN_AADT | AADT in term of natural logarithm | 16,933 | 8.88 | 1.18 | 4.06 | 12.73 |

Note: * denotes categorical variable.

In Table 2, the overdispersion parameter is significant (0.143, SD = 0.002, *t*-statistic > 1.96). Therefore, it can be concluded that the NB fits that Poisson model. The result of Vuong statistic is 18.916 (*t*-statistic > 1.96). It could be stated that the ZINB is proved to be more appropriate. In summary, the study found that the Akaike information criterion (AIC) values continuously decrease until the SZINB reaches the smallest value. Here, a small AIC value indicated a model with better fit [12,47]. However, this study was analyzed based on the spatial analysis model, which used correlation spatial data. Vaida and Blanchard [48] suggested that the conditional Akaike information criterion (cAIC) was a more appropriate model than the ordinary AIC. The penalty term in cAIC is related to the effective degrees of freedom $\hat{\rho}$ [49]. The cAIC allows for the comparison of models with different random effect structures. It likewise allows for the comparison of mixed-effect models with cluster-specific models where the parameters are fixed. The result of the cAIC shows that the SZINB is the smallest. This finding can be interpreted in such a way that the SZINB is the most-fitted model [50].

**Table 2.** Crash frequency model results.

| Model | Log-Likelihood | AIC | cAIC | Pseudo-R$^2$ |
|---|---|---|---|---|
| 1. Poisson regression (Po) | −196,868.2 | 393,758.0 | 393,760.418 | 0.278 |
| 2. Negative binomial (NB) | −30,074.3 | 60,173.0 | 60,172.704 | 0.043 |
| 3. Zero-inflated negative binomial (ZINB) | −29,420.0 | 58,888.5 | 58,886.034 | 0.060 |
| 4. Spatial zero-inflated negative binomial (SZINB) | −29,123.4 | 58,316.9 | 58,304.843 | 0.061 |

Note: ICC = 0.097. Overdispersion parameter ($\alpha$) = 0.143, SD = 0.002 ($p < 0.000$). Vuong z-statistic = 18.916 ($p < 0.000$). The Akaike information criterion (AIC) is calculated as AIC = 2(K − LL(m1)). Where, K is the number of model parameters. LL(m1) and LL(m0) are log-likelihood of model with and without regressor, respectively. The conditional Akaike information criterion (cAIC) is calculated as $cAIC = -2 \times log\text{-}likelihood + 2 \times K$, where K is given by $K = \frac{N(N-p-1)}{(N-p)(N-p-2)}(\hat{\rho}+1) + \frac{N(p+1)}{(N-p)(N-p-2)}$, $p$ is the number of column or the number of fixed effects and $N$ = number of road segments, $\hat{\rho}$ is the effective degree of freedom. Pseudo-R$^2$ is calculated as Pseudo-R$^2$ = 1 − ((LL(m0)/LL(m1)).

The pseudo-R$^2$ of SZINB is relatively low because there are many variables (e.g., human factors, weather, lighting, traffic signs, and land use factors) that are not measured [51]. However, the pseudo-R$^2$ of SZINB was still within the acceptable criteria of Mohammadi et al. [52], Abdul Manan et al. [53]. These authors all accepted the goodness of fit at 0.06.

The model results of SZINB (Table 3) were presented in three parts:

1.  The random effect conditional model refers to the prediction of the relationship between the independent and dependent variables. These variables were grouped according to the sub-areas of the Department of Highway (DOH).
2.  The normal state denotes the factors influencing collision number as non-zero state.
3.  Lastly, zero state pertains to the factors that do not lead to crashes. The zero sate is modeled to find the significant variables that could be specifically led to be the zero-crash segment. These significant variables can be developed to be effective road safety policy.

The inverted signs between states mean that the factors positively or negatively affected the crash frequency. These factors include road length and LN_AADT. It is positive in a normal state and negative in a zero state. Thus, these variables were clearly interpreted to have increased crash frequency. This was expounded in the study of Dong et al. [54].

According to the intercept of random effect, it is statistically significant and different from zero [55]. This could mean that the crash frequencies vary across office highway districts. The variances of random effects for variables of road segment levels were found to be insignificant. In other words, the parameter estimation values were lower than those of the standard errors. Thus, the results indicate that such variables do not differently affect crash frequency in the sub-areas of the DOH. This finding is consistent with the study done by Aguero-Valverde [28]. He found that the random effects of spatial error

were not significant. The fact that the variance of random effect was not significant could be explained by the highway district office having no different control power on distance. As such, the traffic volume and the proportion of trucks were not significantly different in each area under each highway district office's responsibility. Therefore, a management policy that aims to reduce the number of crashes can be uniformly applied throughout the country.

**Table 3.** Parameter estimation of the SZINB model.

| Variable | Normal-State | | | | Zero-State | | | |
|---|---|---|---|---|---|---|---|---|
| | Est. | Std. | *p*-Value | Sig. | Est. | Std. | *p*-Value | Sig. |
| Fixed effect | | | | | | | | |
| (Intercept) | −5.097 | 0.521 | <0.000 | *** | 2.460 | 0.898 | 0.006 | ** |
| length | 0.100 | 0.006 | <0.000 | *** | −1.755 | 0.171 | <0.000 | *** |
| No. of Lane | 0.110 | 0.019 | <0.000 | *** | −0.032 | 0.024 | 0.191 | |
| Concrete (=1) | −0.171 | 0.099 | 0.083 | . | −0.252 | 0.123 | 0.040 | * |
| Lane width | 0.115 | 0.127 | 0.362 | | −0.056 | 0.235 | 0.813 | |
| Footbath (=1) | 0.217 | 0.121 | 0.074 | . | −0.358 | 0.178 | 0.044 | * |
| Shoulder width | 0.226 | 0.035 | <0.000 | *** | 0.002 | 0.053 | 0.968 | |
| Median (=1) | 0.524 | 0.093 | <0.000 | *** | −0.446 | 0.125 | <0.000 | *** |
| Median width | 0.028 | 0.013 | 0.028 | * | 0.026 | 0.017 | 0.116 | |
| LN_AADT | 0.544 | 0.036 | <0.000 | *** | −0.125 | 0.045 | 0.006 | ** |
| Percent of truck | −0.002 | 0.003 | 0.423 | | 0.000 | 0.004 | 0.981 | |
| Random effect: Group = DOH_CODE | | | | | | | | |
| (Intercept) | 2.238 | 1.496 | | | | | | |
| Distance | 0.001 | 0.027 | | | | | | |
| LN AADT | 0.033 | 0.182 | | | | | | |
| Percent of truck | 0.000 | 0.013 | | | | | | |

Note: significant codes: 0 '***'; 0.001 '**'; 0.01 '*'; 0.05 '.'; 0.1.

Traffic volume and the presence of road medians were the top factors that lead to crash occurrences. The reason behind this notion is that these two variables are clearly related. Roads with high volumes of traffic and that are built with medians subsequently lead to increased chances of crashes. Aguero-Valverde [56], Liu et al. [57] provided relevant study results. The reason why the existence of traffic islands caused more crashes could be explained by the fact that the Department of Thai Highways designed roads according to their traffic volume [58]. Roads with traffic islands are usually roads with high traffic volume [57]. Road shoulder and the traffic island width are other factors that contribute to car crashes. These results were inconsistent with those of Joon-Ki et al. [8], which suggested that roads with narrow shoulders led to more frequency of trailing. However, for developing countries, Mahmud et al. [31] found that shoulders did not contribute to more frequent crashes. As for the relationship of these two factors with traffic volume, it was found that the two factors positively correlated with traffic volume (Figure 4). Nevertheless, these two factors should be further investigated to discover why increasing road width of traffic islands and shoulders increased the crash frequency in Thailand.

For zero states, the majority of negative values indicated that significant factors typically lead to an increase in crash frequency. For example, the increasing distance between road segments can lead to the non-zero state, followed by the existence of medians and increase in traffic volume [56]. In view of the intercept of zero states, it was clearly found that it had a significant positive effect on a non-crashes. This implied that the mean frequency of crashes on each road segment was zero. Meanwhile, most of these factors led to a negative effect in zero states, which means an increase in crash frequency.

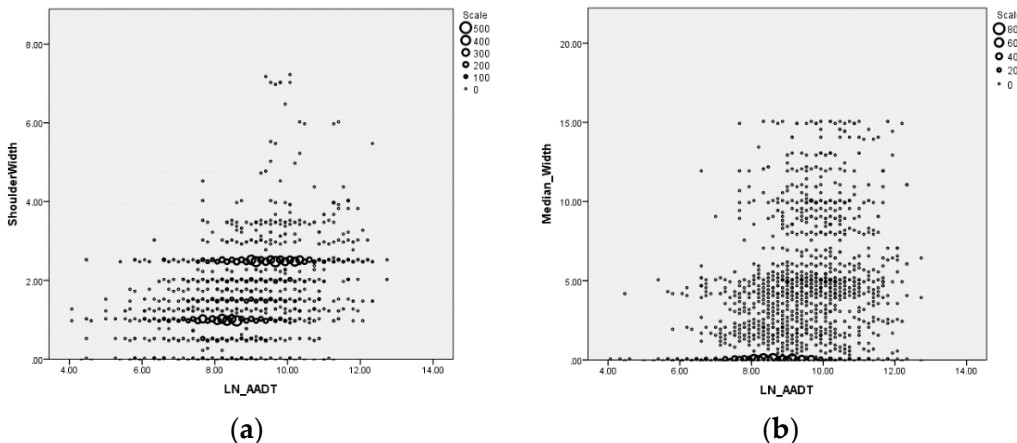

**Figure 4.** Distribution of shoulder width (**a**) and median width (m) (**b**) based on Log AADT.

### 3.2. Crash Severity Model

Table 4 displays the data on crash severity, which indicates the number of road crash victims. After screening, data on 37,685 victims (2014–2018) were collected. Results showed that in terms of driver characteristics, age was mostly a factor for middle-aged people. Moreover, it was shown that most drivers do not use seat belts and helmets. Drunk driving is found to be very rare, which accounts for only 2% of the total number of crashes. For road factors, the study found that the overall picture points to roads with medians of 33%. Median types consist of raised median (26%), depressed median (23%), and colored median (5%). Environmental factors imply that most crashes occur during daytime (60%). On the other hand, nighttime crashes and crashes in poorly illuminated areas reached 10%. In terms of crash-type factor, the study further found that single crashes obtained the highest proportion at 49%. This was followed by rear-end crashes at 25%, side rear-end crashes at 13%, crashes against people traveling, and head-on collisions.

Table 5 provides the parameter estimation results given that frequency data are considered with empirical data. The study found that the mixed-effect model produced more accurate prediction results than the fixed-effect model. This is evidenced by the AIC and pseudo-$R^2$ values. This result is in line with many past studies [19,24,59]. An ICC value of 0.24 can be interpreted as the variability of severity in highway districts at approximately 24%. Thus, the dataset of the current study is suitable for spatial model analysis [19,20].

Table 6 illustrates the comparison of the parameter estimation between traditional logistic regression and the mixed-effect model. The overall numbers of the factors influencing fatal crashes are found along the same direction. In terms of fixed parameters, the mixed-effect model has a smaller number of significant variables. This finding is relevant to that of Yu et al. [47], Wali et al. [60]. Few studies found that the mixed effect model has significant variables greater than or equal to a fixed-effect model [19]. However, there is no study necessarily comparing the number of significant variables between mixed and fixed-effect models. Contradictorily, it was found that the estimation is insignificant. This result is relevant to the study by Ahmed et al. [26], which found that the parameter estimation of the road grade between the Poisson and spatial model (−0.530 and 0.273, respectively) was insignificant.

In terms of vehicle size, the findings show that small cars, such as motor vehicles, incurred the largest fatality risk. This point is understandable due to the physical characteristics of small cars (i.e., mostly motorbikes), which are relatively less secure than medium- and large-sized cars. Thus, high fatality risks are the result [21,22,61]. Similar to many results, females were more likely to die than males. This notion is possibly caused by the females' decision-making phase, which takes more time than that of males [5]. The outstanding result, which indicates that seat belt use can lead to decreased instances of death, is similar to that of Delen et al. [62]. In their study, they found that using seat

belts was the first factor in assessing crash severity. Moreover, drunk drivers were more likely to die than those who did not drink [7]. With regard to the driver's age, the present study found that the elderly were more likely to die. This tendency is consistent with the work of Abdel-Aty [5], who posited that death may be common among the elderly due to their physical condition. They are more prone to injuries than younger drivers [63]. In terms of environmental factors, nighttime collisions were most likely to cause fatality, whereas collisions at night and in poorly illuminated areas increased the likelihood of death [18,64,65].

Road factors pointed to the main traffic lane as a source of low fatality risk [66]. A subsequent variable is the areas where vehicles decelerated before entering. The model results revealed that crashes at median openings and intersections also led to less cases of death. However, rear-end crashes frequently occurred in such areas. An explanation for this finding is that auxiliary lanes are located in areas with median openings. In addition, clear warning signs are likely posted before junctions. Therefore, every car may possibly recognize these areas and slows down. This result is consistent with that of Chen et al. [21].

**Table 4.** Data description of the crash severity model.

| Factor | Descriptions | N | M | SD | MIN | MAX |
|---|---|---|---|---|---|---|
| Fatal injury | 1 = Fatal crash, 0 = Otherwise | 37,685 | 0.14 | 0.35 | 0 | 1 |
| Vehicle size | 1 = Small, 2 = Middle, and 3 = Large | 37,685 | 1.97 | 0.6 | 1 | 3 |
| Driver age | Age of driver | 37,685 | 2.55 | 1.17 | 1 | 7 |
| Gender | 1 = Male, 0 = Female | 37,685 | 0.15 | 0.36 | 0 | 1 |
| Safety equip | 1 = Use, 0 = Otherwise | 37,685 | 0.36 | 0.48 | 0 | 1 |
| Drunk driver | 1 = Yes, 0 = Otherwise | 37,685 | 0.02 | 0.15 | 0 | 1 |
| Main road | 1 = Inner lane, 0 = Otherwise | 37,685 | 0.13 | 0.34 | 0 | 1 |
| Normal | 1 = Normal status *, 0 = Otherwise | 37,685 | 0.97 | 0.17 | 0 | 1 |
| Divided road | 1 = Divided road, 0 = Otherwise | 37,685 | 0.33 | 0.47 | 0 | 1 |
| Flush | 1 = Flush median, 0 = Otherwise | 37,685 | 0.05 | 0.21 | 0 | 1 |
| Raised | 1 = Raised median, 0 = Otherwise | 37,685 | 0.26 | 0.44 | 0 | 1 |
| Depressed | 1 = Depressed median, 0 = Otherwise | 37,685 | 0.23 | 0.42 | 0 | 1 |
| Barrier | 1 = Yes, 0 = Otherwise | 37,685 | 0.12 | 0.33 | 0 | 1 |
| Concrete | 1 = Yes, 0 = Otherwise | 37,685 | 0.11 | 0.32 | 0 | 1 |
| Straight | 1 = Yes, 0 = Otherwise | 37,685 | 0.84 | 0.37 | 0 | 1 |
| Slope | 1 = Yes, 0 = Otherwise | 37,685 | 0.06 | 0.24 | 0 | 1 |
| Intersection | 1 = Yes, 0 = Otherwise | 37,685 | 0.14 | 0.35 | 0 | 1 |
| Median opening | 1 = Yes, 0 = Otherwise | 37,685 | 0.10 | 0.3 | 0 | 1 |
| Road surfaces | 1 = Dry, 0 = Otherwise | 37,685 | 0.12 | 0.33 | 0 | 1 |
| Weather | 1 = Clean, 0 = Otherwise | 37,685 | 0.13 | 0.33 | 0 | 1 |
| Day | 1 = Yes, 0 = Otherwise | 37,685 | 0.60 | 0.49 | 0 | 1 |
| Darkness | 1 = Nighttime and non-lighting, 0 = Otherwise | 37,685 | 0.10 | 0.29 | 0 | 1 |
| Pedestrians | 1 = Pedestrian crash, 0 = Otherwise | 37,685 | 0.07 | 0.26 | 0 | 1 |
| Rear-end | 1 = Rear-end crash, 0 = Otherwise | 37,685 | 0.25 | 0.43 | 0 | 1 |
| Sideswipe | 1 = Sideswipe crash, 0 = Otherwise | 37,685 | 0.13 | 0.34 | 0 | 1 |
| Single vehicle | 1 = Single-vehicle crash, 0 = Otherwise | 37,685 | 0.41 | 0.49 | 0 | 1 |
| Head-on | 1 = Head-on crash, 0 = Otherwise | 37,685 | 0.03 | 0.16 | 0 | 1 |
| Other | 1 = Other crash type, 0 = Otherwise | 37,685 | 0.08 | 0.27 | 0 | 1 |

Note: * Normal status means that the road is not undergoing maintenance.

**Table 5.** Goodness of fit for the crash severity models.

| Value/Model | Traditional Model | Random Parameter Model |
|---|---|---|
| Log-likelihood | −13,642.11 | −11,652.1 |
| AIC | 27,341.97 | 23,436.01 |
| McFadden $R^2$ | 0.101 | 0.149 |

Note: Intra-class correlation coefficient (ICC) = 0.240.

**Table 6.** Parameter estimation of the fatal crash model.

| Variables | Traditional Model | | | Random Effect Model | | |
|---|---|---|---|---|---|---|
| | Estimate | Std. Error | *p*-Value | Estimate | Std. Error | *p*-Value |
| Fixed effect | | | | | | |
| (Intercept) | 0.550 | 0.177 | 0.002 | 0.292 | 0.304 | 0.337 |
| Vehicle size (=2) | −0.655 | 0.039 | <0.000 | −1.115 | 0.058 | <0.000 |
| Vehicle size (=3) | −0.550 | 0.052 | <0.000 | −1.169 | 0.075 | <0.000 |
| Gender (=1) | −0.359 | 0.048 | <0.000 | −0.422 | 0.067 | <0.000 |
| Safety equip (=1) | −0.538 | 0.036 | <0.000 | −0.512 | 0.061 | <0.000 |
| Drunk driver (=1) | 0.316 | 0.088 | <0.000 | 0.306 | 0.123 | 0.036 |
| Driver age | 0.087 | 0.013 | <0.000 | 0.090 | 0.018 | <0.000 |
| Day (=1) | −0.223 | 0.037 | <0.000 | −0.317 | 0.053 | <0.000 |
| Darkness (=1) | 0.653 | 0.051 | <0.000 | 0.552 | 0.074 | <0.000 |
| Road surfaces (=1) | 0.000 | 0.107 | 0.997 | −0.029 | 0.147 | 0.842 |
| Weather (=1) | −0.076 | 0.104 | 0.462 | −0.170 | 0.143 | 0.234 |
| Normal (=1) | −0.116 | 0.091 | 0.202 | −0.088 | 0.149 | 0.553 |
| Main Road (=1) | −0.184 | 0.065 | 0.005 | −0.288 | 0.104 | 0.006 |
| Divided road (=1) | −0.586 | 0.135 | <0.000 | −0.343 | 0.240 | 0.153 |
| No. of lane | −0.099 | 0.011 | <0.000 | −0.055 | 0.020 | 0.007 |
| Flush (=1) | −0.338 | 0.147 | 0.021 | −0.488 | 0.303 | 0.108 |
| Raised (=1) | −0.729 | 0.137 | <0.000 | −0.612 | 0.254 | 0.016 |
| Depressed (=1) | −0.469 | 0.137 | 0.001 | −0.167 | 0.247 | 0.499 |
| Barrier (=1) | −1.028 | 0.153 | <0.000 | −0.637 | 0.270 | 0.018 |
| Intersection (=1) | −0.307 | 0.047 | <0.000 | −2.050 | 0.308 | <0.000 |
| Median opening (=1) | 0.217 | 0.054 | <0.000 | −0.152 | 0.095 | 0.110 |
| Straight (=1) | −0.150 | 0.050 | 0.003 | −0.102 | 0.069 | 0.144 |
| Concrete (=1) | −0.182 | 0.058 | 0.002 | −0.073 | 0.108 | 0.497 |
| Slope (=1) | 0.183 | 0.068 | 0.007 | 0.171 | 0.098 | 0.082 |
| Pedestrians (=1) | −0.634 | 0.076 | <0.000 | −0.158 | 0.235 | 0.874 |
| Rear-end (=1) | −0.353 | 0.048 | <0.000 | −0.481 | 0.126 | 0.001 |
| Sideswipe (=1) | −0.667 | 0.058 | <0.000 | −2.528 | 0.483 | <0.000 |
| Single vehicle (=1) | −1.050 | 0.049 | <0.000 | −1.182 | 0.098 | <0.000 |
| Head-on (=1) | 0.509 | 0.079 | <0.000 | 0.397 | 0.444 | 0.536 |
| | | | | Variance | Std. Error | *p*-Value |
| Random effect | | | | | | |
| Intercept | | | | 0.018 | 0.135 | 0.791 |
| Divided Median (=1) | | | | 1.434 | 1.197 | 0.389 |
| Flush (=1) | | | | 1.437 | 1.199 | 0.389 |
| Median opening (=1) | | | | 5.109 | 2.260 | 0.062 |
| Intersection (=1) | | | | 10.680 | 3.268 | 0.004 |
| Raised (=1) | | | | 2.525 | 1.589 | 0.226 |
| Depressed (=1) | | | | 0.407 | 0.638 | 0.651 |
| Pedestrians (=1) | | | | 2.182 | 1.477 | 0.268 |
| Rear-end (=1) | | | | 4.280 | 2.069 | 0.094 |
| Sideswipe (=1) | | | | 18.320 | 4.281 | <0.000 |
| Single vehicle (=1) | | | | 2.631 | 1.622 | 0.214 |
| Head-on (=1) | | | | 21.540 | 4.642 | <0.000 |

Lastly, in terms of crash types, the parameter estimation still goes on the same side in both the traditional and random effect model. According to the random effect model, there were two types of collisions in general. These are the pedestrian and head-on crashes. These two types of crashes had non-significant effects on the fixed coefficient. In the random effect, the pedestrian crash had not been found to be significant, while a head-on crash was significant, which implied that when a crash pattern was determined to vary along the road segment, there would be a significant effect on crash fatality. This is relevant to the study by Zeng et al. [63] who found that head-on crashes are significant on the variance of a random term. In Thailand, head-on crashes are slightly more dangerous. This is because in some districts (e.g., industrial zones, university zones), small vehicles are

often reversing. For sideswipe crashes, it was found that they tended to have a significant effect on non-fatal collisions. Such finding was inconsistent with that of Chen et al. [21]. Nevertheless, when considering the constant values from the model of Xiao et al. [40], it was found that sideswipe collisions (same direction) negatively affected fatality crashes. Meanwhile, rear-end crashes were often the collisions that resulted in no fatalities. These results were consistent with the findings of Kim et al. [19]. They found that when the intercept was compared with crash fatality among crash types, it yielded the most negative values of rear-end crashes. In other words, the average rear-end crashes would cause the least fatality rates. The reason for this is that rear-end crashes often occurred frequently, especially in an urban area where vehicle speed was not different. This resulted in only few instances of death. The studies concluded that rear-end collisions in urban areas were less violent than those in rural areas [42]. That is, a single crash led to fewer instances of death. Moreover, if there were no fixed objects, the likelihood of death was also lower [65].

The variance of random effect was found to be insignificant at the 0.05 level (i.e., intercept, meaning that the mean chance of fatal crashes was not different between road segments under the control of the highway district office and the other variables such as the divided median, flush median, raised median, depressed median, and median opening). This implied that the relationship between these factors and crash fatality did not vary with road segments [12]. However, many variables did (i.e., intersection and sideswipe crashes). This meant that each crash at an intersection in each road segment was different [55]. The causes of the difference may come from the control power of the highway district office over most of the urban roads. This directly affected the speed of vehicles and may lead to different levels of severity of intersection crashes. The variance of head-on collisions was also significant, indicating that the roads under the control of the highway district office caused different severity levels of head-on crashes. The results depended on different types of traffic islands. These results suggested that barriers should be built by highway district offices who have the control over several rural roadways.

## 4. Conclusions and Implementation

The study aimed to focus on the factors influencing the frequency and severity of road traffic crashes. The abovementioned factors were divided into two levels: collision case and highways. These factors were analyzed using the spatial model. The model results predicted crash risks for road segments under the responsibility of the highway districts.

For the crash frequency model, a Vuong z-statistic test found that a zero-inflated model is more appropriate than an NB model. According to the ICC, it was found that 9.7% of the crash frequency varies correspondingly with the responding area of the Department of Highway district (DOH). Moreover, the SZINB is the most suitable based on the conditional cAIC. According to the results of the SZINB, in terms of a fixed parameter that consists of a normal state and a zero state, these states are consistent. Meanwhile, if a factor is positive in the normal state, that factor would be negative in the zero state. For example, road length and traffic volume (i.e., for the random effect model, it was found that only the intercept is significant) could be interpreted as the crash frequency had unobserved heterogeneities. Thus, spatial analysis should be applied to analyze the crash frequency data.

The crash severity model made an analysis based on the logistic regression models for understanding the relationship between the contributing factors and crash fatality (1 = fatal crash, 0 = non-fatal crash). A comparison between the fixed-effect and mixed-effect models was made. This was indicated by the AIC and pseudo-$R^2$. It was found that the mixed-effect model is fit. Besides, based on the ICC of crash fatality, there was a 24% variance among road segments. Thus, the crash severity model should be accounted for by the unobserved heterogeneity model or a mixed effect based on the spatial model. The results of the fixed coefficient estimation between the two models are a consistent sign. The number of significant parameters in the mixed-effect model is fewer. The parameter estimation of random coefficients indicated many significant factors. These factors affected crash fatality varying among the road segments (e.g., intersection, sideswipe collision,

and head-on collision). The overall fixed coefficient estimation makes sense. For example, a small vehicle crashing may more likely lead to a fatal crash than a large vehicle. Moreover, a crash at night, without lights, increased chances of a fatal crash. The study subsequently proposed policies that target the decrease in the crash frequency and fatal crashes.

The results confirmed that applying spatial correlation to the sub-areas of the DOH is more suitable than the traditional model in terms of crash frequency and severity in Thailand. The proposed guidelines are summarized as follows:

The DOH can implement uniform policies throughout the country with special consideration for roads with median and high traffic volumes. In the case of such roads, a road safety audit may be implemented for a safety assessment. Authorities responsible for road supervision should regularly inspect the safety of such roads as an extra safety measure. For example, road safety audit (RSA) and identification of hazardous locations (IHL) should be thoroughly implemented to improve road safety in the case of limited funding. Road safety policy in Thailand with regards to RSA and IHL is not routine performed. When the DOH wants to perform RSA or IHL, they will hire a consultancy company. Thus, the RSA or IHL will proceed a few years at a time. Previous research about road safety in Thailand used to suggest a policy about RSA and IHL such as Jomnonkwao et al. [61], Champahom et al. [67], Se et al. [68]. However, they suggest only generally but did not specify. A new policy from the result of this study suggests the DOH should begin to do the RSA and IHL specifically on the road segment with the median with high AADT and more than four lanes road. Those road segments usually connect between large areas such as provinces or districts, since the fixed effect of AADT, median, and number of lanes are significant. Moreover, the random effect confirms significant variance among the department of highways offices.

Guidance policy reduces crash fatalities, which is selected based on significant parameters from the model. The guidances include four parts, as follows:

- A survey for poorly lighted crash-prone areas should be first conducted. There is previous research that suggested this point such as Champahom et al. [42], Se et al. [68], Se et al. [69]. These works mentioned that the DOH should survey the dark road segments and consider installing lighting poles. According to the road safety policy in Thailand, there are regularly surveys of lighting conditions on the road segments. However, some processes are slow and limited in terms of budget. To conserve the budget for installing lighting-poles, this study recommends surveying a light condition of more than 3% at the slope segment first, since the factors significantly positively affect likelihood of fatality.
- Followed by the posting of additional signs for anti-drunk driving campaigns, to reduce fatal crashes. Law enforcement and a campaign against drunk drivers is a regular proceeding, mentioned in previous works, namely, Jomnonkwao et al. [70], Se et al. [71]. To establish the guidelines for determining campaigns developed by Phillips et al. [72], healthy organizations such as the Thai Health Promotion Foundation should use personal communication or roadside media as part of their campaign delivery strategy. Lastly, checkpoints inspecting for alcohol use and seat belt use should likewise be reinforced. Recommendation of using safety equipment has been mentioned in terms of drivers' attitudes [73–75] and in terms of law enforcement by using CCTV for detection [76,77].
- The Department of Land Transport can help promote such campaigns and topic training because head-to-head collisions and rear-end crashes are the most frequently occurring crash types that lead to high rates of fatality. Otherwise, short-term campaigns can be used as guidelines [72]. Therefore, it is necessary to consider areas prone to crashes, especially where driving on the wrong side of the road is a likely tendency. In Thailand, the campaign or the training topic of awareness of collision types are a low priority. The previous research recommended that the DOH should highlight some crash types such as rear-end crashes, single crashes and head-on crashes [42,67,68]. These studies suggest some policies, for example, the driver should be aware of the appropriate headway with the front vehicle. Regarding the results

of this study, rear-end crashes and head-on collisions were found to have significant variance among the road segments. Thus, a new policy from this study recommended that the road safety government does not necessarily attempt implementation of safety guidelines throughout the country. However, they should focus on some areas, for example, roads with mixed vehicle types, such as industrial areas and communities without limited accessibility of trucks.

- In addition, officials of the DOH should investigate the blind spots and risk spots, especially undivided roads and roads divided by flush medians. Currently, the safety policy of Thailand is trying to reduce the length of flush medians and to increase the length of barriers. It is relevant to the road safety studies in Thailand, which suggest reducing flush medians because it is more likely to sustain fatal injuries in a crash [42,69]. This study confirms significant spatial variance of median-openings and intersections. Therefore, the recommendation of this study is to begin considering installation of barriers to divide intersections and median opening points.

## 5. Limitations and Further Research

This study has its limitations. First, the independent variables have rarely been introduced into the crash frequency model due to the limited data in Thailand. Thus, the policies for reducing crash rates should be interpreted with caution. This is important, especially as there are other developing countries that will apply this idea in the future. Moreover, the pseudo-$R^2$ seems low because the relative factors were not included in the model. This study could not be fully accounted for due to the limitation in the dataset used. Other variables, such as the ratio of curves and the number of intersections, should also be considered [78].

**Author Contributions:** Conceptualization, T.C.; data curation, T.C. and W.N.; formal analysis, T.C. and S.J.; funding acquisition, V.R. and A.K.; investigation, T.C. and C.B.; methodology, T.C.; resources, S.J.; supervision, V.R. and A.K.; validation, T.C. and W.N.; writing—original draft, T.C. and C.B.; writing—review and editing, T.C. All authors have read and agreed to the published version of the manuscript.

**Funding:** This work was supported by the Suranaree University of Technology (SUT) and the Thailand Science Research and Innovation (TSRI) (grant number: RSA6280061 and Fulltime Full-time 61/22/2563), and the Mitsui Sumitomo Insurance Welfare Foundation (MSIWF). The APC was funded by the Suranaree University of Technology.

**Institutional Review Board Statement:** The study was conducted according to the guidelines of the Declaration of Helsinki, and approved by the Ethics Committee of Suranaree University of Technology (COA.95/2562, 16 December 2019).

**Informed Consent Statement:** Not applicable.

**Data Availability Statement:** Data are available on request due to privacy restrictions.

**Acknowledgments:** The authors would like to thank Enago (www.enago.com, accessed on 9 January 2005) for the English language review.

**Conflicts of Interest:** The authors declare no conflict of interest.

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
