# Peer review of "Analysis of Crash Frequency and Crash Severity in Thailand: Hierarchical Structure Models Approach"

_sustainability, doi:10.3390/su131810086_

Round 1

Reviewer 1 Report

A great paper with several important findings on Thai highway crashes, I have following minor revisions:

  1. The authors often mentioned crashes as accidents. Please use crashes instead of accidents throughout the paper.
  2. In page 10, the authors modeled both normal-state and zero-state. Please justify the reasons of modeling zero-state in a few sentences.
  3. In page 12, any reason of having mixed effect model less significant? 
  4. Before developing the models, did the authors checked whether the explanatory variables were correlated?  Please specify. 
  5. In table 4, please explain what you meant by normal status. 

Author Response

Response to Reviewer 1

A great paper with several important findings on Thai highway crashes, I have following minor revisions:

1. The authors often mentioned crashes as accidents. Please use crashes instead of accidents throughout the paper.

Response 1: We thank you for your suggestion. We have revised and used only “crashes” consistently throughout the manuscript.

2. In page 10, the authors modeled both normal-state and zero-state. Please justify the reasons of modeling zero-state in a few sentences.

Response 2: We thank you for your suggestion. We have added the justification of zero-state model in line 349 as follow:

             The zero sate is modeled to find the significant variables that could be specifically led to be the zero-crash segment. These significant variables can be developed to be the effective road safety policy.

3. In page 12, any reason of having mixed effect model less significant? 

Response 3: We thank you for your question. This sentence seems confusing. The term “less significant” should be “a smaller number of significant variables”. However, there is no study investigated why the mixed effect model generates fewer significant variables than the traditional model. We have revised it already in line 430 as follow,

            In terms of fixed parameters in the mixed-effect model has a smaller number of significant variables. This finding is relevant to that of Wali et al. (2018); Yu et al. (2019).  Few studies found that the mixed effect model has significant variables greater than or equal to a fixed-effect model (Kim et al., 2007). However, there is no study necessarily comparing the number of significant variables between mixed and fixed-effect model.

4. Before developing the models, did the authors checked whether the explanatory variables were correlated?  Please specify. 

Response 4: We thank you for your comment. Before developing the model, the correlations of explanatory variables have been checked. There is no pair greater than  0.7 which could be multi-collinearity problem. We have explained in the manuscript in line 288 as follow:

            Before developing the model, the correlations between explanatory variables have been checked. It was found that there is no pair has correlation greater than 0.7, which was an appropriate indicator when collinearity begins to severely distort model estimation and subsequent prediction (Dormann et al., 2013).

5. In table 4, please explain what you meant by normal status. 

Response 5: We thank you for your suggestion. We have added the note of table 4 to explain, as follow:

            Note: * Normal Status mean that the road is not under maintenance

Reviewer 2 Report

The paper provides a statistical analysis of the Crash Frequency and Crash Fatalities for the whole country for Thailand. Overall the paper is within the scope of Sustainability for its potential implications in improving road-safety policies in Thailand. There are major limitations in the current version of the manuscript which may require substantial improvements.

The recommendations are very generic and it is not clear what is new. It would be useful to review relevant previous studies on this topic in Thailand, as well as review  existing government policies to improve road safety in Thailand. Authors could  then elaborate on the recommendations and clarify what is new in this study. 

Author Response

Response to Reviewer 2

            The paper provides a statistical analysis of the Crash Frequency and Crash Fatalities for the whole country for Thailand. Overall the paper is within the scope of Sustainability for its potential implications in improving road-safety policies in Thailand. There are major limitations in the current version of the manuscript which may require substantial improvements.

1. The recommendations are very generic and it is not clear what is new. It would be useful to review relevant previous studies on this topic in Thailand, as well as review existing government policies to improve road safety in Thailand. Authors could  then elaborate on the recommendations and clarify what is new in this study. 

            Response 1: We thank you for your suggestion. We have revised and added discussion to our manuscript in term of the following points:

  • We have provided the specific recommendations.
  • We have explained clearly which recommendations are developed from the result of this study.
  • We have provided the previous studies and government policies in Thailand.

The revised recommendation is as follow:

The results confirmed that applying spatial correlation to the sub-areas of the DOH is more suitable than the traditional model in terms of crash frequency and severity in Thailand. The proposed guidelines are summarized as follows:

The DOH can implement uniform policies throughout the country with special consideration for roads with median and high traffic volumes. In the case of such roads, a road safety audit may be implemented for a safety assessment. Authorities responsible for road supervision should regularly inspect the safety of such roads as an extra safety measure. For example, Road safety audit (RSA) and identification of hazardous locations (IHL) should be thoroughly implemented to improve road safety in the case of limited funding. Road safety policy in Thailand about RSA and IHL is not routine works. When the DOH want to do RSA or IHL, they will hire a consultancy company. Thus, the RSA or IHL will proceed few years per time. The previous research about road safety in Thailand used to suggest a policy about RSA and IHL such as Champahom et al. (2019); Jomnonkwao et al. (2020); Se et al. (2020a). However, they suggest only overall but did not specify. A new policy from the result of this study suggests the DOH should begin to do the RSA and IHL specifically on the road segment with the median with high AADT and more than 4 lanes road. Those road segments usually are connecting between large areas such as provinces or districts, since the fixed effect of AADT, Median and Number of lanes are significant. Moreover, the random effect confirms significant variance among locations of the department of highways offices.

Guidance policy reduces the crash fatality, which is selected based on significant parameters from the model. The guidances include four parts, as follows:

1) A survey for poorly lighted crash-prone areas should be first conducted. There is previous research that suggested this point such as Champahom et al. (2020); Se et al. (2021a); Se et al. (2021b). These works mentioned that the DOH should survey the dark road segment and consider installing the lighting poles. Accounting to the road safety policy in Thailand, there are regularly surveying lighting conditions on the road segment. However, some process is a slow and limited in term of budget. To save the budget for installing lighting-pole, this study recommends surveying a light condition at the slope segment more than 3% first. Since the factors positively affect likelihood of fatality, significantly.

2) followed by the posting of additional signs for anti-drunk driving campaigns, to reduce fatal crashes. Law enforcement and a campaign for the drunk driver is a regular proceeding. Some previous work mentioned, for example, Se et al. (2020b); Jomnonkwao et al. (2021). To establish the guidelines for determining campaigns developed by Phillips et al. (2011), healthy organizations such as the Thai Health Promotion Foundation should use personal communication or roadside media as part of their campaign delivery strategy. Lastly, checkpoints inspecting for alcohol use and seat belt use should likewise be reinforced. Recommendation of using safety equipment has been mentioned such as in terms of drivers’ attitude  (Ratanavaraha & Jomnonkwao, 2013; Nambulee et al., 2018; Nambulee et al., 2019) and terms of law enforcement by using CCTVs’ detection (Kumphong et al., 2018; Satiennam et al., 2020).

3) The Department of Land Transport can help promote such campaigns and topic training because head-to-head collisions and rear-end crashes are the most frequently occurring crash types that lead to high rates of fatality. Otherwise, short-term campaigns can be used as guidelines  (Phillips et al., 2011). Therefore, it is necessary to consider crash-prone areas, especially where driving on the wrong side of the road is a likely tendency. In Thailand, the campaign or the training topic of awareness of collision types are a low priority. The previous research recemented that DOH should highlight some crash types such as rear-end crash, single crash and head-on crash (Champahom et al., 2019; Champahom et al., 2020; Se et al., 2021b). These studies suggest some policies, for example, the driver should aware of the appropriate headway with the front vehicle. Regrind the results of this study, rear-end crash and head-on collision were found significant variance among the road segments. Thus, a new policy from this study recommended that the road safety government does not necessarily attempt throughout the country. But they should focus on some areas, for example, the road with mixed vehicle types, such as industrial areas and the community without limited accessibility of the truck.

4) In addition, officials of the DOH should investigate the blind spots and risk spots, especially the undivided road and divided road by flush median. Currently, The safety policy of Thailand is trying to reduce the length of the flush median and increase the length of the barrier. It is relevant to the road safety studies in Thailand, which suggest reducing flush median because it is more likely to sustain fatal injury in a crash (Champahom et al., 2020; Se et al., 2021a). This study confirms significant spatial variance of median-opening and intersection. Therefore, the recommendation from this study is to begin considering installation of the barrier to divide the intersection and median opening point.

Reviewer 3 Report

This study focuses on crash frequency on road segments and crashes severity in Thailand. That is, this study aims to understand the crash frequency and fatality.

The authors analyzed a significant number of factors that could impact the crash frequency on the road segments and the crash severity.

They conducted a comprehensive statistical analysis and presented the analysis results and a discussion in two parts: the crash frequency model and crash severity model.

The obtained results were compared with the results of other authors, and for those results that differed, they gave an appropriate explanation (e.g., the specifics of driver behaviour in Thailand on some road segments).

They identified which of the analyzed factors contributed the most to the frequency of the collision and its severity.

They identified which of the analyzed factors contributed the most to the crash frequency and its severity.

Based on the obtained results, they made specific recommendations for various stakeholders whose measures can affect the reduction of the crash frequency and their severity.

They provided the limitations of the study.

There are minor errors in the paper:

  • An inappropriate font is used (lines 196, 204, 235, 240, and 433);
  • The sentence (s) (lines 223-225) require(s) revision and correction;
  • The order of the labels in the index (Xijm) is not the same as in the formula (2) (line 196);
  • Line 250: explanation of the labels from the formula (?0? and ?0? indicate…)
  • Line 338: pseudo-R2 -> pseudo-R2.

Author Response

Response to Reviewer 3

1. An inappropriate font is used (lines 196, 204, 235, 240, and 433);

Response 1: We thank you for your comment. We have correctly revised them already.

2. The sentence (s) (lines 223-225) require(s) revision and correction;

 Response 2: We thank you for your comment. We have revised as follows:

            In other words, various characteristics of the areas have indifferently affected the likelihood of fatality, which determines the two severity levels including the fatal and non−fatal crashes.

3. The order of the labels in the index (Xijm) is not the same as in the formula (2) (line 196);

Response 3: We thank you for your comment. We have now revised them. The index and equation are same.          

4. Line 250: explanation of the labels from the formula (?0? and ?0? indicate…)

            Response 4: We thank you for your comment. We have revise as follows:

 indicate the slope or relationship with  corresponding to  and , respectively.

5. Line 338: pseudo-R2 -> pseudo-R2.

            Response 5: We thank you for your comment. We have now correctly revised them.

Round 2

Reviewer 2 Report

I appreciate the responses from the authors.

Author Response

Response: Thank you very much for your valuable comments. English has been checked throughout the manuscript.